# Mechanisms for Business Ecosystem Members to Capture Value through the Strong Network Effect

**Haruo Awano * and Masaharu Tsujimoto**

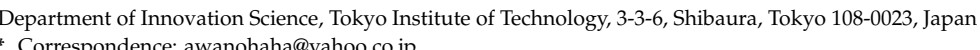

Department of Innovation Science, Tokyo Institute of Technology, 3-3-6, Shibaura, Tokyo 108-0023, Japan
* Correspondence: awanohaha@yahoo.co.jp

**Abstract:** Business ecosystem leaders tend to focus on their own success rather than carefully monitoring the success of business ecosystem members, and each member must find a mechanism to capture value. Rarely has business ecosystem research examined the success of business ecosystem members. This study investigates the mechanisms by which business ecosystem members capture value in the business ecosystem related to different types of storage formats and how these mechanisms contribute to the sustainability of the business ecosystem. We identify two value capture mechanisms in the storage business ecosystem. First, media manufacturers, being members of the storage business ecosystem, can capture value through strong network effect in the server backup markets. They can sell a significant amount of media for every single unit of a drive sold through the strong network effect. Second, media manufacturers can capture value by selling their media repeatedly as consumables for many years. We quantitatively confirm these mechanisms using a vector autoregressive model to measure the strength of the indirect network effect. These mechanisms help business ecosystem members in staying within the business ecosystem and also contribute to the business ecosystem's sustainability.

**Keywords:** business ecosystem; sustainability; network effect; ecosystem member; complementor; vector autoregressive model; storage format

## 1. Introduction

In the era of big data and the Internet of Things (IoT), a business ecosystem has become critical to a company's success. Business leaders are often more concerned with their own success than with the success of other members of the business ecosystem [1]. Conversely, a business ecosystem leader's success is dependent on other members of the ecosystem [1]. All business ecosystem members and leaders are potentially dependent on one another through multilateral dependencies [1,2]. The success of a business ecosystem is determined by the actions of self-interested actors who join the network; thus, attracting members to a business ecosystem necessitates an understanding of what motivates the potential members, particularly how participation relates to achieving their specific goals [1]. Although members of a business ecosystem typically strive to advance the ecosystem's success, they tend to prioritize their self-interest [1]. Each business ecosystem member must devise a mechanism to capture value beyond what is contributed to the business ecosystem's joint value generation [1,3]. Business ecosystem research has rarely investigated the success of the ecosystem members [1].

It is critical for both business ecosystem leaders and members to understand how to develop a sustainable business ecosystem. Network effect is crucial for an ecosystem's leader and members to capture value. Measurement of variation in network effects can be regarded as a key future research challenge [4].

This study, inspired by the insights of existing literature, focuses on the following questions: what mechanisms enable business ecosystem members to capture value from the business ecosystem? How do these mechanisms contribute to the sustainability of a

business ecosystem? How can network effects be measured for a business ecosystem? By addressing these questions, this study intends to complement previous research, thereby contributing to existing literature.

Storage technology and products are important in the era of big data. To store an ever-increasing amount of data, different types of storage formats have been developed. For example, the 1/2 inch cartridge format, linear tape open (LTO) format, digital linear tape (DLT) format, advanced intelligent tape (AIT) format, digital data storage (DDS) format, VXAformat, Travan format and 90 mm Magneto-optical (MO) have been introduced to meet market demands for capacity, transfer speed, and cost. Drive manufacturers such as International Business Machines Corp (IBM), (Tempe, AZ, USA), Hewlett-Packard (HP), (Bristol, UK), Quantum (San Jose, CA, USA), Sony drive company (Tokyo, Japan), Ecrix (Boulder, CO, USA), 3M Company (Saint Paul, MN, USA), Fujitsu (Kanagawa, Japan), and Olympus (Tokyo, Japan) are ecosystem leaders, whereas media manufacturers such as Fujifilm (Tokyo, Japan), Hitachi Maxell (Tokyo, Japan), Sony media company (Tokyo, Japan), 3M, TDK (Tokyo, Japan), Imation (Oakdale, MN, USA), and so on are ecosystem members. Drive manufacturers and media manufacturers form the business ecosystem for each storage format. These drive manufacturers and media manufacturers engage in extensive research and development (R&D) to enhance the formats' storage capacity to create new generations of the formats. This R&D ensures the life of the formats and enhances its sustainability.

The purpose of this research is to investigate how media manufacturers capture value from the business ecosystem as its members while contributing to the business ecosystem by manufacturing and selling complementary media products. The problem is that little research exists to study what mechanisms enable business ecosystem members to capture value from the business ecosystem [1]. If business ecosystem members cannot find the mechanism to capture value from the business ecosystem, they either exit the business ecosystem or move to other business ecosystems [5–7]. Then, the business ecosystem loses sustainability. Therefore, we also aim to examine how the value capture mechanisms contribute to the business ecosystem's sustainability. Furthermore, we aim to confirm the mechanism by evaluating the strength of network effects with a vector autoregressive (VAR) model. Various storage format cases have been studied in this research.

The remainder of this paper is organized as follows. Section 2 discusses the related literature. Section 3 describes the research methods employed in the study. Section 4 analyzes the value capture mechanisms for business ecosystem members. Section 5 discusses the empirical findings and presents our conclusions and offers some practical and theoretical implications, as well as limitations.

## 2. Literature Review

### 2.1. Business Ecosystem

The concept of business ecosystem is gaining traction in the field of technology and innovation management [8]. A business ecosystem is newly defined as an interdependent network of self-interested actors jointly creating value [1]. Business ecosystem is a descriptive term that generally explains today's interconnected business [9]. Business ecosystem simply means the network of organizations participating in a common business [9]. The competition among enterprises is no longer the competition between individual enterprises but based on the co-evolution under the business ecosystem [10].

Previous studies on ecosystems can be categorized into four research streams. Studies using the business ecosystem perspective focus on the business context and set value capture and/or value creation as central variables [8]. These studies aimed to reveal the dynamics and patterns of ecosystems and organizational behavior [8]. The researchers of business ecosystem focused on business player networks and analyzed the mechanisms behind the networks [8]. At most basic level, a network is defined by the structure of the relationships among the ecosystem members and a business ecosystem leader [1,11]. The business model theory was clarified to describe the characteristics of the ecosystem

concept [1,8,12,13]. In conventional business model theory, the focus is on how a single organization creates value, influenced by economic and institutional factors. In the case of a business ecosystem, interactions between different business ecosystem members take place to create value for end-users/customers [14]. An ecosystem perspective has become increasingly important with platform-based business models enabled by a modular division of labor [1,2,15].The effect on further ecosystem innovation will depend on the platform ecosystem's decision about the degree of openness of its digital interface [16]. Ecosystems provide an important option for firms to harness external partners for value creation as part of an open innovation strategy [1,17].

A business ecosystem has two aspects. First, the company needs to build a business ecosystem to realize continual product update and iteration to fulfil the expanding customer demands [10]. Second, all business ecosystems experience a loop of creation, expansion, and extinction [10]. Business ecosystems expand and fall not just as a result of the lifecycle of technological products but also as a result of the lifecycle of the ecosystems themselves [5]. For example, a piece of video game hardware, being a technological system, loses its superiority or edge when new technology or hardware is introduced. However, because platform adoption is dependent on both complementors and consumers, the ecosystem's expansion and decline are not necessarily dependent on the platform's technological supremacy [5–7].

Modern businesses must strengthen their own competencies; more importantly, they need to be supported by a flexible and stable business ecosystem [10]. A business ecosystem leader is defined as the organization(s) that, through some combination of resources, leadership and control, hold a unique position to both support and benefit from ecosystem success [1]. A business ecosystem leader needs to adjust its strategy based on the performance of other firms in the ecosystem [1,18]. Because of technological interdependence, the competitive advantage of business ecosystem leaders depends on their components from suppliers and complements from complementors [19]. Business ecosystem governance by ecosystem leaders is studied regarding the indirect control of ecosystem members' incremental and radical behaviors in goods development and successful innovations [20] Business ecosystem leaders cannot directly control business ecosystem members similarly to how firms control subsidiaries [2,20]. Not only the business ecosystem leader but also the business ecosystem members change their roles across phases of business ecosystem development [1,21]. The multi partner nature of the business ecosystem was focused on to explain the dynamics among business ecosystem members and a business ecosystem leader [1,22].

A business ecosystem leader's success is dependent on the success of other members that are complementors of business ecosystem members [1]. Certain complementors that contribute to the symbiotic co-existence within a platform ecosystem could merge as keystone firms/companies in the same business ecosystem [5]. They can contribute to the sustainability of platform-based markets and facilitate the co-existence of multiple platform ecosystems [5].

Three broad groups of papers are identified for the study regarding ecosystem [2]. A business ecosystem stream centers on a firm and its environment. An innovation ecosystem stream focuses around a particular innovation or new value proposition and the constellation of actors that support it. A platform ecosystem stream considers how actors organize around a platform [2]. A platform ecosystem is studied regarding various themes such as takeoff and emergence [23–25], servitization [26], cluster [27], legitimation [28], persistence [29], decentralized governance [30,31] and competition within complementary product markets [32]. All actors in platform ecosystems have participated in both value creation and capture within the boundaries of a particular ecosystem. That is, they have competed for their share of value capture according to mutually enforced rules and governance schemes as well as individual complementors' ability to attract users and generate revenue [33]. Some research highlights the specific characteristics of business ecosystem members, such as the participation of complementors as partners in co-creating

value [1,34]. Entrepreneurial ecosystem stream [1,35–37], and regional ecosystem stream are also investigated [1,38].

### 2.2. Sustainability

Sustainability is perceived to generate value for consumers, businesses, and stakeholders [39]. Business ecosystem sustainability has emerged as the primary issue that enterprises must address [10]. Even an innovative and technologically superior business ecosystem cannot be sustained if the business ecosystem members are not successful in developing and providing complementary goods [5]. Therefore, a business ecosystem leader must attract and retain high-quality innovators in the ecosystem to ensure sustainable development [40]. A business ecosystem leader should not only protect its own survival, but also support the system's health and stability to ensure its own as well as the ecosystem's sustainability [10].

A company can compete with its competitors through differentiation. When there are multiple generations of a product, the new technology for each generation can be differentiated. As a result, the performance of each new technology generation is, therefore, superior to that offered by the old technology [41]. In this manner, firms achieve sustainability with each new generation of products that are developed to meet the needs of the firms' existing customers, who continuously demand improved performance at a lower cost [41].

The entry of high-quality complementors as a significant factor in ecosystem sustainability is investigated here [40]. This research stream is expanded in terms of "unsustainability caused by physical intermediary firms" and "maintaining sustainability by introducing an ecosystem strategy" [42].

### 2.3. Value Capture

A linkage of capabilities interacting among ecosystem members and an ecosystem leader contributes to creating an ecosystem-level advantage [1,43]. However, how business ecosystem members can capture the value by monetizing it in some way becomes an open question in a technologically mediated world, where regulation provides some loose contours of how sectors can work and what actors can legitimately sell [2,44]. The very things that make it easy to capture value within an ecosystem make it harder to recruit business ecosystem members [2]. Business ecosystem members may decide to shift to another ecosystem if the conditions no longer favor them [2]. Most business ecosystem members are complementors with very limited power [2,45]. While aligning with the business ecosystem rules, each business ecosystem member must decide whether to specialize in a given business ecosystem or across multiple business ecosystems to capture values [46].

### 2.4. Network Effect

The network effect has its origins in the development of telephone services at the beginning of the 20th century. It was then popularized by the invention of Ethernet and, for many years, was associated with telecommunication, information and IT infrastructure [47] The gain received from a product or service and the number of users mutually and cyclically increase when such a mechanism works; thereby, the product or service is spread exponentially [6]. The network effect is defined as the phenomenon in which the benefit of using a product or service increases when the number of users who use the same or compatible products or services increases [47]. The increased value accrued by network participants is contingent on the number of other users in the network with whom they can interact [4,48]. Business ecosystems are reinforced by mutual dependency [4] as the number of customers joining a network increase [48]. Indirect network effects occur when the benefits from using a product improve with the use of a complementary set of compatible goods [47]. Indirect network effects are ancillary benefits that accrue to network participants as the network grows, such as the development of complementary services, standards formation, and price reduction [30].

How can strategy researchers empirically measure such variance in network effect influence [4]? One possibility in the search for effective measures of the intensity of the network effects is to look at analogous measures [4]. Social network analysis measures that account for the degree, symmetry, and strength of network ties among participants may provide a more detailed picture of network intensity in a specific situation [4]. In this study, we examine the strengths of network effects by quantitatively validating them using a VAR model.

The network effect can be used to solve important social problems. How the network effect concept can be applied to the social business ecosystem can be an object for future research.

## 3. Materials and Methods

### 3.1. Data

Both primary and secondary data sources are used for this study. One of the authors has worked with Sony on strategic planning, R&D, marketing, production, business alliances, and new product and technology planning for storage media, such as LTO, DLT, DDS, AIT, VXA, Travan, and 90 mm MO media. For more than 30 years, this author worked for Sony Japan and Sony USA as a general manager, manager, engineer, and in-house attorney. This author reviewed his experience of storage media and derived the knowledge regarding storage media business such as sales & marketing, R&D and production for this study. We used this knowledge to determine the characteristics of sales & marketing, R&D and production for storage media business. The knowledge derived by the author contributed to qualitative analysis in Section 4.1. The result of this qualitative analysis contributed by the author became the base for statistical analysis in Section 4.2.

The storage business cases were studied alongside a review of third-party resources. To acquire objective and quantitative statistics, published materials from the 2011 Techno Systems Research Co., Ltd. market report were analyzed. The 2011 Techno Systems Research Co., Ltd. storage market report, titled "Digital Media/Storage Outlook For 2018," provides the number of drives and media sold from 2001 to 2011 [49]. Market data were extrapolated for periods when no reported information was available. The market data, such as the number of drives and media of the storage market report, are used for statistical analysis in Sections 4.2 and 5.1. The reason why the data from 2001 to 2011 are used in this study is that drives and media of various kinds of storage formats were sold during this period from 2001 to 2011. We can compare the business of these storage formats for the same period of years and can expect to obtain useful implications.

### 3.2. The Storage Business: Types of Storage Formats and Tape Library

In 2000, IBM, HP, and Quantum (formerly Seagate) commercialized an open format of LTO, which has become the industry standard of the midrange class after winning the competition with digital linear tape (DLT) and advanced intelligent tape (AIT). DLT was developed in 1984 by Digital Equipment Corporation (DEC), which was later purchased by Quantum Corp, and the DLT tape drive was manufactured by Quantum and Tandberg. Sony developed and manufactured the AIT format, which was a helical scan tape drive. Sony and HP pioneered the digital data storage (DDS) format in 1989 by employing the digital audio tape (DAT) cassette. Ecrix developed the VXA format in 1999, and Exabyte produced the VXA tape drives following the merger of Ecrix and Exabyte. IBM developed the 1/2-inch cartridge format in 1984, which had a half-inch tape spooled into a data cartridge holding a single reel. This was a high-end storage tape format used for server backup and data archiving. While LTO, DLT, and AIT were midrange formats, DDS and VXA were low-end ones.

Fujitsu and IBM, among others, introduced 90 mm Magneto-optical (MO) in 1991. The 90 mm MO format was designed for the consumer market, such as personal computer (PC). In 1995, 3M introduced the Travan format, an 8 mm magnetic tape drive that was also aimed at the consumer market.

A tape library is a storage device that includes one or more tape drives, a number of slots for tape media, and an automated loading system. Since it holds many tape media ranging from eight to hundreds, the tape library can store a significant amount of data. A tape library's target market is server backup and data archiving.

### 3.3. Variables

The number of media, our objective variable, is the number of media of a certain storage format sold in a given year. The number of drives of a certain storage format sold in a given year is our explanatory variable.

### 3.4. Statistical Analysis

We employ the VAR model for our empirical analysis to confirm the results of our qualitative study about the factor for the number of media sold in a given year. The number of media of format $i$ sold in a given year $t$ can be obtained using pth-order VAR (VAR(y)) model as follows:

$$q_{im}(t) = c + a_1 q_{id}(t) + a_2 q_{id}(t-1) + a_3 q_{id}(t-2) + \ldots + a_p q_{id}(t-y) + \varepsilon_i$$

where $q_{im}(t)$ is the number of media of format $i$ sold in year $t$; $q_{id}(t)$ is the number of drives of format $i$ sold in year $t$; and $\varepsilon_i$ is the error term. $q_{im}(t)$ is a linear combination of the number of drives sold in the last $y$ years. We assume $a_1 = a_2 = a_3 = \ldots = a_p = a$ and estimate the following equation:

$$q_{im}(t) = c + a(q_{id}(t) + q_{id}(t-1) + q_{id}(t-2) + \ldots + q_{id}(t-y)) + \varepsilon_i \qquad (1)$$

where $(q_{id}(t) + q_{id}(t-1) + q_{id}(t-2) + \ldots \ldots + q_{id}(t-y))$ is the cumulative number of drives sold over the last $y$ years including the year $t$. $a$ is the intensity of indirect network effect per a drive. The value of $a$ can be determined through regression analysis. Furthermore, the value of "y" can be determined by finding the lag value that maximizes adjusted R-squared.

## 4. Results

### 4.1. Qualitative Analysis

We used the knowledge described in Section 3.1. and we found the following characteristics of sales & marketing, R&D and production for storage media business. As complementors, media manufacturers such as Fujifilm, Sony, Hitachi Maxell, and others could sell their multiple media for the sale of a single drive. They could also continue to sell media for drives sold in the past. For years, they could receive repeated orders from their customers. This is primarily because the media business is a consumables business. For customers who needed to store a large amount of data, there was a tape library with multiple media. Media manufacturers could sell a significant amount of media for tape library drives. This could have been an incentive for media manufacturers as business ecosystem members to stay in the storage formats ecosystem and contribute to the ecosystem's sustainability. Media manufacturers had to actively engage in cost reduction to ensure that customers could purchase multiple media at a reasonable price. Media manufacturers developed new media technologies for the next generation of storage formats so that the format could continue to meet customer needs. This is the mechanism by which media manufacturers and business ecosystem members could continue to capture value in the business ecosystem. This mechanism also contributed to the business ecosystem's sustainability.

### 4.2. Statistical Analysis

Using the VAR model shown in Equation (1), we quantitatively confirmed the results derived in Section 4.1. Furthermore, we determined the value of $a$ through regression analysis and the value of $y$ by calculating the lag value that maximizes adjusted R-squared.

In the case of the 1/2-inch cartridge format, we calculated Equation (1) through regression analysis with changing lag value of y from three years to nine years. We have

found that the adjusted R-squared monotonically increases from three years to seven years and monotonically decreases from seven years to nine years. The adjusted R-squared is maximized at seven years. Therefore, we determine the value of y is seven years in the case of the 1/2-inch cartridge format. Then the value of *a* is determined from the calculation when we adopt y = seven years. We used the data of the number of drives and media of the market report described in Section 3.1 [49]. We similarly calculated Equation (1) through regression analysis in the case of other formats such as LTO format, DLT format, etc. Table 1 presents the statistical analysis results for the 1/2-inch cartridge format. The number of media sold in a given year is the objective variable. The cumulative number of drives has a Pr value of less than 0.05. Therefore, the number of media sold in a year is statistically significantly affected by the cumulative number of drives sold over the previous seven years, including that year. Furthermore, the intensity of indirect network effect per drive is found to be 81.02. This implies that the number of media sold in a year is approximately 81 per drive sold over the previous seven years, including that year. The media have been repeatedly ordered for seven years.

**Table 1.** Statistical analysis for the 1/2-inch cartridge format.

|  | Coefficient "a" | Pr |
| --- | --- | --- |
| (Intercept) | −2543 | $2.79 \times 10^{-3}$ |
| Cumulative number of drives | 81.02 | $2.00 \times 10^{-5}$ |

Note. Adjusted R-squared = 0.9757, *y* = 7 years.

Table 2 presents the statiscal analysis results for the LTO format. The number of media sold in a given year is the objective variable. The cumulative number of drives has a Pr value of less than 0.05. Therefore, the number of media sold in a year is statistically significantly affected by the cumulative number of drives sold over the previous two years, including that year. Furthermore, the intensity of indirect network effect per drive is found to be 24.31. This implies that the number of media sold in a year is about 24 per drive sold in the previous two years, including that year. The media has been repeatedly ordered for two years.

**Table 2.** Statistical analysis for the LTO format.

|  | Coefficient "a" | Pr |
| --- | --- | --- |
| (Intercept) | −1069.7 | $1.44 \times 10^{-1}$ |
| Cumulative number of drives | 24.314 | $1.28 \times 10^{-8}$ |

Note. Adjusted R-squared = 0.9910, *y* = 2 years.

Table 3 presents the statitical analysis results for the DLT format. The number of media sold in a given year is the objective variable. The cumulative number of drives has a Pr value of less than 0.05. Therefore, the number of media sold in a year is statistically significantly affected by the cumulative number of drives sold over the previous five years, including that year. The intensity of indirect network effect per drive is found to be 7.03. This implies that the number of media sold in a year is about seven per drive sold in the previous five years, including that year. The media have been repeatedly ordered for five years.

**Table 3.** Statistical analysis for the DLT format.

|  | Coefficient "a" | Pr |
| --- | --- | --- |
| (Intercept) | 475.8 | $1.81 \times 10^{-1}$ |
| Cumulative number of drives | 7.03 | $6.50 \times 10^{-8}$ |

Note. Adjusted R-squared = 0.9856, *y* = 5 years.

Table 4 presents the statistical analysis results for the DDS format. The number of media sold in a given year is the objective variable. The cumulative number of drives has a Pr value of less than 0.05. Therefore, the number of media sold in a year is statistically significantly affected by the cumulative number of drives sold over the previous three years, including that year. The intensity of indirect network effect per drive is found to be 9.493. This implies that the number of media sold in a year is about nine per drive sold in the previous three years, including that year. The media have been repeatedly ordered for three years.

**Table 4.** Statistical analysis for the DDS format.

|  | Coefficient "a" | Pr |
|---|---|---|
| (Intercept) | −3352.1 | $2.48 \times 10^{-3}$ |
| Cumulative number of drives | 9.493 | $4.00 \times 10^{-9}$ |

Note. Adjusted R-squared = 0.9935, $y$ years = 3 years.

Table 5 presents the statistical analysis results for the AIT format. The number of media sold in a given year is the objective variable. The cumulative number of drives has a Pr value of less than 0.05. Therefore, the number of media sold in a year is statistically significantly affected by the cumulative number of drives sold over the previous three years, including that year. The intensity of indirect network effect per drive is found to be 6.321. This means that the number of media sold in a year is about six per drive sold in the previous three years, including that year. The media have been repeatedly ordered for three years.

**Table 5.** Statistical analysis for the AIT format.

|  | Coefficient "a" | Pr |
|---|---|---|
| (Intercept) | 19.04 | $9.49 \times 10^{-1}$ |
| Cumulative number of drives | 6.321 | $1.50 \times 10^{-4}$ |

Note. Adjusted R-squared = 0.8703, $y$ years = 3 years.

Table 6 presents the statistical analysis results for the VXA format. The number of media sold in a given year is the objective variable. The cumulative number of drives has a Pr value of less than 0.05. Therefore, the number of media sold in a year is statistically significantly affected by the cumulative number of drives sold over the previous two years, including that year. The intensity of indirect network effect per drive is found to be 9.995. This implies that the number of media sold in a year is about ten per drive sold in the previous two years, including that year. The media have been repeatedly ordered for two years.

**Table 6.** Statistical analysis for the VXA format.

|  | Coefficient "a" | Pr |
|---|---|---|
| (Intercept) | −33.12 | $5.41 \times 10^{-1}$ |
| Cumulative number of drives | 9.995 | $2.90 \times 10^{-5}$ |

Note. Adjusted R-squared = 0.9185, $y$ = 2 years.

The above statistical analysis results for different types of storage formats show that the intensity of indirect network effect per drive ranges from 6.321 to 81.02. For the 1/2-inch cartridge format and LTO format, the number of media sold per drive are approximately 81 and 24, repectively. Media manufacturers as business ecosystem members could sell a significant amount of media for the sale of one drive. Furthermore, we quantitatively confirm that media were repeatedly ordered for a period between two and seven years.

## 5. Discussion and Conclusions

To summarize, we have discovered mechanisms that allow members of a business ecosystem to capture value through network effect. We have quantitatively confirmed the mechanisms using the VAR model shown in Equation (1). One mechanism is that media manufacturers, business ecosystem members can sell a significant amount of media for the sale of a single drive through network effect in the business ecosystem and earn profits. Another mechanism is that media manufacturers can sell their media repeatedly as consumables for many years and earn profits. The purpose of a business is to generate profitable and sustainable revenue streams [47]. The aforementioned mechanisms contribute to the generation of a sustainable business.

Members of a business ecosystem develop and commercialize complementary products, which are one of the main points of interest for customers [40]. Media manufacturers must reduce the cost of media, allowing customers to buy large amounts of media at reasonable prices for many years. Furthermore, it is critical that media manufacturers develop new complementary technology for the next generation of storage formats so that customers can continue to buy drives and media for many years.

How to measure the difference in network effect is mentioned as an important future research question [4]. In this study, we have determined the strength of the indirect network effect (denoted *a*) using regression analysis. We have found that the intensity of indirect network effect of 1/2 inch cartridge format and LTO format is larger than those of DLT, AIT, DDS and VXA. It is logical that in markets with stronger network effects, installed base size will have a greater impact on future growth; however, such size-on-growth dynamics are more likely a result of strong network effects than a measure of their intensity [4].

### 5.1. Network Effect Strength: A Comparative Analysis of the Travan and 90 mm MO Formats

To compare the strength of the network effect, we conduct statistical analysis using the VAR model presented in Equation (1) for the Travan and 90 mm MO formats. Both these formats were aimed at the consumer market, such as the PC. Conversely, 1/2-inch cartridge, LTO, DLT, AIT, DDS, and VXA formats were aimed at the high end, midrange, and low end of server backup markets.

Table 7 presents the statistical analysis results for the Travan format. The number of media sold in a given year is the objective variable. The cumulative number of drives has a Pr value of less than 0.05. Therefore, the number of media sold in a year is statistically significantly affected by the cumulative number of drives sold over the previous three years, including that year. The intensity of indirect network effect per drive is found to be 2.1635. This implies that the number of Travan media sold in a year is about two per drive sold in the previous three years, including that year. Manufacturers of Travan-format media could sell approximately two media for the sale of one drive. This is significantly less than the intensity of indirect effect observed for the 1/2-inch cartridge, LTO, DLT, AIT, DDS, and VXA formats. Specifically, the intensity of indirect network effect observed for the Travan format is an order of magnitude smaller than that of the 1/2-inch cartridge and LTO formats. The Travan media were repeatedly ordered for three years.

**Table 7.** Statistical analysis for the Travan format.

|  | Coefficient "a" | Pr |
|---|---|---|
| (Intercept) | 185.53 | $6.23 \times 10^{-2}$ |
| Cumulative number of drives | 2.1635 | $3.90 \times 10^{-8}$ |

Note. Adjusted R-squared = 0.98755, $y$ = 3 years.

Table 8 presents the statistical analysis results for the 90 mm MO format. The number of media sold in a given year is the objective variable. The cumulative number of drives has a Pr value of less than 0.05. Therefore, the number of media sold in a year is statistically significantly affected by the cumulative number of drives sold over the previous five years,

including that year. The intensity of indirect network effect per drive is found to be 2.5818. This implies that the number of 90 mm MO media sold in a year is approximately three per drive sold in the previous five years, including that year. This is significantly less than the intensity of indirect effect observed for the 1/2-inch cartridge, LTO, DLT, AIT, DDS, and VXA formats. Specifically, the intensity of indirect network effect observed for the 90 mm MO format is an order of magnitude smaller than that of the 1/2-inch cartridge and LTO formats. The 90 mm MO media were repeatedly ordered for five years.

**Table 8.** Statistical analysis for the 90 mm MO format.

|  | Coefficient "a" | Pr |
| --- | --- | --- |
| (Intercept) | 1571.64 | $2.76 \times 10^{-2}$ |
| Cumulative number of drives | 2.5818 | $5.94 \times 10^{-6}$ |

Note. Adjusted R-squared = 0.98502, $y$ = 5 years.

The statistical analysis results for the Travan and 90 mm MO formats show that the intensity of indirect network effect of 1/2-inch cartridge, LTO, DLT, AIT, DDS, and VXA for the server backup market is significantly higher than that of the Travan and 90 mm MO formats for the PC backup market. It is noteworthy that the intensity of indirect network effect of the 1/2-inch cartridge format for the high-end backup market and the LTO format for the midrange backup market are an order of magnitude larger than those of the Travan and 90 mm MO formats for the PC backup market. Future research must examine the factors that drive the strength of network effects and how they manifest differently across markets [4]. The first driving force in the storage industry is that server backup necessitates more data storage space than PC backup. The tape library used for high end and midrange backups consume a large amount of media, which is the second driving force in the storage industry.

*5.2. Theoretical Implications*

This research provides several theoretical implications for the study of business ecosystem. First, we have found the mechanism that allow business ecosystem members of media manufacturers to capture value by strong network effect in the server backup markets. They can obtain profits by strong network effects. Research on business ecosystems has rarely examined the success of business ecosystem members apart from business ecosystem leaders [1]. This paper complements these previous studies by investigating business ecosystem members' mechanisms to capture value by strong network effect.

Second, our paper contributes to the literature on the sustainability of the business ecosystem. Previous studies have observed that when members of a business ecosystem who serve as complementors fail to earn profits, they either exit the business ecosystem or move to other business ecosystems [5–7]. One reason for the loss of sustainability in business ecosystems is the profitability of business ecosystem members [50]. Business ecosystem members would not be motivated to remain with the ecosystem at low profit [50]. The mechanism we have found for a business ecosystem member to capture value is by using a strong network effect to help business ecosystem members to stay within the business ecosystem. This mechanism contributes to the sustainability of the business ecosystem. Our findings support the theory that each business ecosystem member must devise mechanisms to capture value beyond its contribution to the ecosystem's joint value creation [1].

Third, our research contributes to the related literature on the network effect. How to measure the network effect is mentioned as an important future research question [4]. This study shows that the regression analysis using a VAR model can be applied to determine the strength of the indirect network effect for storage business.

*5.3. Practical Implications*

This research provides practical implications for business ecosystem members. Our finding can serve as a reference guide for managers and decision makers, seeking to

capture value when they join a storage business ecosystem as complementors such as media manufacturers. The managers of business ecosystem members should study how they can increase the strength of indirect network effect. First, they should decrease the cost of their media to keep profit margins. Then, they can decrease the selling price so that customers can buy large amounts of media at reasonable prices when they buy storage drives.

Second, the managers should collaborate with the manufacturers of drive-related products, such as a tape library, to promote these products. This increases the strength of the network effect and the sales amount of complements such as storage media.

The managers of business ecosystem members should study how they can continue to conduct sustainable business. They should develop new complementary technology for the next generation of storage formats so that customers can continue to buy drives and media for many years.

### 5.4. Limitations and Future Research

Our research provides critical findings; however, it has certain limitations. First, the business ecosystem leaders and business ecosystem members analyzed in this study are primarily large companies, such as IBM, HP, Fujifilm, and Sony. Our study includes only a single startup, called Ecrix which developed the VXA format as explained in Section 3.2. SMEs, owing to their limited R&D budget, typically cannot spend considerable funds on the development of new complementary products for the next generation of storage formats. Further studies, therefore, must investigate SMEs and more startups to further validate our results.

Second, this study examines the business ecosystems related to different types of storage formats. As a result, our findings are affected by the nature of the products in the storage industry. Both the business ecosystem leaders and business ecosystem members in this study are hardware product manufacturers. Since software products have not been analyzed, this study may not accurately reflect the situation of the software industry, such as video game software or complementary products of the video game business. Moreover, software products such as PC operating systems and smartphone applications have not been analyzed in this study. Therefore, future research should focus on whether the results of our study are applicable to industries other than storage products.

Third, our qualitative investigation of how value capture mechanisms contribute to a business ecosystem's sustainability limits our understanding of the sustainability concept. It is mentioned that unclear understanding of sustainability can be included in the limitations for the study about sustainability [51]. Sustainability is a critical issue for the leader and members of a business ecosystem. Future research, therefore, should quantitively investigate how value capture mechanisms contribute to a business ecosystem's sustainability. This can help scholars better understand sustainability.

Furthermore, future research must investigate the business areas, other than storage, in which we can apply the VAR model to measure the strength of the indirect network effect.

**Author Contributions:** Conceptualization, H.A. and M.T.; methodology, H.A. and M.T.; formal analysis, H.A.; investigation, H.A.; data curation, H.A.; writing—original draft preparation, H.A.; writing—review and editing, H.A. and M.T.; supervision, M.T. All authors have read and agreed to the published version of the manuscript.

**Funding:** This research received no external funding.

**Institutional Review Board Statement:** Not applicable.

**Informed Consent Statement:** Not applicable.

**Data Availability Statement:** Data available in a publicly accessible repository.

**Conflicts of Interest:** The authors declare no conflict of interest.

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
