# Peer review of "Mechanisms for Business Ecosystem Members to Capture Value through the Strong Network Effect"

_sustainability, doi:10.3390/su141811595_

Round 1

Reviewer 1 Report

Dear authors,

Your article needs to be strengthened in terms of innovation and conclusion.

Complete the following corrections:

1- Replace all the references under the year 2015 with the last 2 years.

2- The author has self-cited in 4 references, which are 10% references, which is unethical work.

3- Strengthen the problem and statement of the problem and point it out exactly.

4- When stating the problem, point out what your issue has to do with the concept of sustainability.

Yours sincerely,

Reviewer 2 Report

Hope my comments/suggestions can assist the authors in improving the manuscript. 

INTRODUCTION

·            The first three paragraphs of the introduction section were well-written. However, I'm having trouble connecting the discussion in the fourth, fifth, and sixth paragraphs on page 2 with the rest of the discussion. Please revise your discussion to improve the flow of the ideas.

 IMPLICATIONS

·            Section 6.2. Only one sentence to express the implications? This is largely insufficient. The authors should provide more information to justify the study's unique contribution, including both theoretical and practical implications. Both of these implications should be articulated in separate sections titled "theoretical implications" and "practical implications."

Reviewer 3 Report

Dear Author/s

Thank you for the opportunity to read your study that attempts to research mechanisms for business ecosystem members to capture value through network effect. 

The proposed topic is very interesting and addresses current issues business ecosystem modeling. There is a need for papers that propose a new approach to exploring knowledge in this area and map possibilities for the future. The research approach based on vector autoregression model proposed in this paper is undoubtedly in line with the latest research trends.

The structure of the work is an example of developing the subject of interest by the Author/s.

However, in its current stage, the manuscript can still be improved to be more valuable to the readers. Therefore, I encourage you to make some changes that I believe will be made this paper stronger. Please see my detailed comments below. Good luck with your research!

I have provided my comments as follows:

Title –I found it interesting, informative and encouraging to read.

Abstract – The key elements are included.

Keywords - I consider key words such as "value", "profit", "statistical analysis" to be redundant. The authors do not present the results of research on "profit". The keyword "value" is too general, and so is the phrase "statistical analysis" (moreover, "vector autoregressive model" has already been pointed out). 

Introduction - In my opinion, the introduction introduces the reader to the topic under discussion. Due to the fact that the authors have separated a separate section devoted to the literature review, I consider the introduction sufficient. 

Section 2 – Literature review The Section 2  - Literature review - distinguishes 4 subsections:  "Business Ecosystem", "Sustainability", "Value Capture" and "Network Effect".

However, they were prepared on the basis of relatively old literature. Of the 42 references listed in the paper, as many as 17 are older than 10 years.

In my opinion, the article would be stronger if the authors referred to the latest research findings from academic literature.

In conclusion, I encourage a strong rebuilding and completion of this part of the paper.

Section 3

Subsection 3.1. The information in this section is incomprehensible and unclear to me. The experience of one of the authors in the analyzed industry is indicated, but it is not reflected in the rest of the work - it is not clear what knowledge, in what way and in which part of the research this author contributed - how it affected the results, how it was studied, measured, etc.

They further cited a report that is not shown in the references, I could not find a link where the data of this report can be consulted, which exact data the authors used in their study, etc. The authors wrote that the report deals with data from 2001-2011 - why do the authors use such old data? I found no justification or explanation for this. Due to the lack of a reliable description of the data, it is not possible to reliably verify and evaluate the presented results. In my opinion, the paper will be stronger if the authors improve this section by describing both the data and the entire research process in details. In my opinion,  the current presentation of the research process methodology is not clear enough to allow other researchers to replicate the study. I encourage you to improve this part of the paper. I believe that this would be very valuable for further studies on this subject. I encourage you to rename this section to Materials and Methods.

Section 4 - The information in Section 4 is suggested to be included in Section 3. 

Section 5

Due to ambiguities in the methodology section, it is difficult for me to assess the validity of the findings presented in this section.

For example, the authors point to the results of the qualitative analysis in subsection 5.1. Where did these results come from? how was this analysis conducted? what was analyzed? etc. The lack of methodological information unimpossesses the evaluation of the records presented.  Also, the results of the VAR analysis I cannot comment on and evaluate due to the failure to follow the scientific regime of describing the methodological part of the study.  I also encourage a more detailed description of the results of this analysis and not just the end result. In my opinion, the paper will be stronger if the authors improve this section by describing results of research in details.

Section 6

Discussion and conclusion are organized in a way that isn’t easy to understand. This is partly due to previous observations. Surprising, for example, is the sentence "Our study includes only a single startup, called Ecrix". This has not been shown explicitly before. What was the subject of the study, after all?

References

The paper makes use of insufficiently up-to-date literature. The literature includes only 42 items (of which as many as 17 are publications presenting the state of knowledge more than 10 years ago): in the case of a scientific article, in my opinion, this is insufficient to confirm a solid review of the literature and preparation of the author for research.

It was a pleasure to read this manuscript. There is a lot of potential in this paper to make an interesting contrubution to the field. Overall, I find the idea of the study very valuable. However, I encourage you to improve the article so that its content does not raise questions about the scientific regime.  I hope you find the above observations useful as you continue to further develop your study. Good luck with your paper!

Round 2

Reviewer 1 Report

Some bugs are still not fixed:

1- All self-references of the authors should be removed because it is unrelated to the content of the research. This is completely unethical.

2- All the references under the year 2015 should be replaced with the last 2 years.

Reviewer 3 Report

The authors did a great job revising their article. Their article has gained new value after the changes. I accept all changes made by the authors.

Once again, it was a pleasure to read this manuscript. Congratulations. I encourage the authors to further research in this area.

Round 3

Reviewer 1 Report

Thanks for making the corrections.